# Carbapenemase-Producing *Klebsiella pneumoniae* Colonization and Infection in Solid Organ Transplant Recipients: A Single-Center, Retrospective Study

**DOI:** 10.3390/microorganisms9112272

**Published:** 2021-10-31

**Authors:** Nicole Pagani, Silvia Corcione, Tommaso Lupia, Silvia Scabini, Claudia Filippini, Roberto Angilletta, Nour Shbaklo, Simone Mornese Pinna, Renato Romagnoli, Luigi Biancone, Rossana Cavallo, Giovanni Di Perri, Paolo Solidoro, Massimo Boffini, Francesco Giuseppe De Rosa

**Affiliations:** 1St Stephen’s Centre, Chelsea and Westminster Hospital, 252 Fulham Rd., London SW10 9NA, UK; nicolepagani.ao@gmail.com (N.P.); giovanni.diperri@unito.it (G.D.P.); 2Department of Medical Sciences, Infectious Diseases, University of Turin, 10100 Turin, Italy; silvia.corcione@unito.it (S.C.); silvia.scabini@unito.it (S.S.); roberto.angilletta@unito.it (R.A.); nour.shbaklo@edu.unito.it (N.S.); simone.mornesepinna@unito.it (S.M.P.); francescogiuseppe.derosa@unito.it (F.G.D.R.); 3Department of Medical Sciences, Tufts University School of Medicine, Boston, MA 02109, USA; 4Infectious Disease Unit, Cardinal Massaia Hospital, 14100 Asti, Italy; 5Clinical Statistics, Department of Surgical Sciences, University of Torino, 10126 Torino, Italy; claudia.filippini@unito.it; 6General Surgery 2U, Liver Transplant Center, AOU Città Della Salute e Della Scienza di Torino, University of Turin, 10100 Turin, Italy; renato.romagnoli@unito.it; 7S.C. Nefrologia, Dialisi e Trapianto U, AOU Città Della Salute e Della Scienza, 10100 Turin, Italy; luigi.biancone@unito.it; 8Microbiology and Virology Unit, University of Turin, 10100 Turin, Italy; rossana.cavallo@unito.it; 9Cardiovascular and Thoracic Department, Pneumology Unit U, AOU Città Della Salute e Della Scienza di Torino, 10100 Turin, Italy; paolo.solidoro@unito.it; 10Division of Cardiac Surgery, Department of Surgical Sciences, Città Della Salute e Della Scienza, University of Turin, 10100 Turin, Italy; massimo.boffini@unito.it

**Keywords:** carbapenemase, *Klebsiella pneumoniae*, solid organ transplant, colonization

## Abstract

Carbapenemase-KPC producing *Klebsiella pneumoniae* (CP-Kp) infection represents a serious threat to solid organ transplant (SOT). All patients admitted between 1 May 2011 and 31 August 2014 undergoing SOT were included in the retrospective study. The primary outcomes included a description of the association of enteric colonization and invasive infections by CP-*Kp* with one-year mortality. Secondary outcomes were the study of risk factors for colonization and invasive infections by CP-*Kp*. Results: A total of 5.4% (45/828) of SOT recipients had at least one positive rectal swab for CP-*Kp*, with most (88.9%) occurring after transplantation. 4.5% (35/828) of patients developed a CP-*Kp*-related invasive infection, with 68.6% (24/35) being previously colonized. The 1-year mortality was 31.1% in patients with enteric colonization with CP-Kp and, it was 51.4% among patients with CP-*Kp*-related invasive infections. At univariate analysis, colonization, invasive infections, sepsis, severe sepsis, and septic shock were significantly associated with 1-year mortality. At multivariate analysis, only invasive infections and the combination of sepsis, severe sepsis, or septic shock were significantly associated with 1-year mortality, whereas gastrointestinal colonization was significantly associated with survival. In this population, the 1-year mortality was significantly associated with invasive infections; otherwise, gastrointestinal colonization was not associated with increased 1-year mortality.

## 1. Introduction

In recent years, the growing spread of multi-drug-resistant (MDR) organisms has threatened the efficacy of certain advanced care options, such as solid organ transplant (SOT), for a number of end-stage organ diseases by undermining the control of such procedures’ infective complications [1]. Recent studies have shown an increased proportion of Gram-negative bacteria, particularly carbapenemase-producing *Klebsiella pneumoniae* (CP-*Kp*), with high morbidity and mortality [2,3,4]. There are methodological, microbiological, and clinical difficulties in detailing the profound effects of such molecular mechanisms of resistance within the local epidemiology [5]. Furthermore, strategies for managing CP-*Kp* that is colonized in patients undergoing transplantation have not been systematically evaluated (e.g., deferring the procedure, the use of selective intestinal decontamination, peri-operative prophylaxis, or short-term antimicrobial treatment).

These trends strongly highlight the need for sound, effective diagnostic tools for detection and treatment strategies for management of MDR pathogens in SOT recipients to minimize infectious complications and graft failure as well as reduce infection control failures of hospital programs [3,4]. SOT recipients are especially at risk of infection by MDR bacteria. End-stage organ failure, immunosuppression, hospital admissions, surgeries, and invasive medical devices are the main risk factors for infections events in SOT recipients [6]. Transplantation has been identified as an independent risk factor for CP-*Kp* infection along with pneumonia and renal failure [7]

In this retrospective observational study, we aimed to describe the prevalence of gastroenteric colonization and invasive infections that are caused by CP-*Kp* in a cohort of SOT recipients in a large tertiary care hospital in Turin, Italy. We focused our analysis on the impact of CP-*Kp* colonization and invasive infections on mortality one year after transplant.

## 2. Materials and Methods

All patients admitted to the Città della Salute e della Scienza di Torino–Presidio Molinette Hospital between 1 May 2011 and 31 August 2014 undergoing SOT were included in the study. The SOTs included heart, lung, liver, kidney, pancreas, and combined organ transplants that were performed at our institution.

A retrospective review of medical charts was performed to collect demographics and clinical data. These included comorbidities, such as cardiovascular diseases, chronic obstructive pulmonary disease (COPD), chronic kidney failure, chronic liver failure, previous solid and hematological malignancy, splenectomy, previous SOT, *cytomegalovirus* (CMV) serology, hospital admission >48 h within 6 months prior to the SOT, previous isolation of extended-spectrum beta-lactamases (ESBL) or CP bacteria from any clinical specimen within 3 months prior to the SOT. Other data included date of admission and SOT, indication for SOT, date of last post-transplant follow-up visit or death, date of graft loss (where applicable), pre-SOT ICU admission duration, corticosteroid treatment before SOT (10 mg of prednisolone or equivalent per day for ≥7 days pre-SOT), complications after SOT, which were defined as hemodynamic instability, acute kidney injury (AKI), sepsis, severe sepsis or septic shock [8], mechanical ventilation or extracorporeal membrane oxygenation (ECMO), re-transplant, other unplanned surgery after SOT, CMV primary infection or reactivation and immunosuppressive treatment. Data on age, sex, ICU stay duration prior to organ removal, CMV serology, and bacterial isolates on blood, urine, and respiratory tract samples taken on the day of the organ removal were collected for each organ donor. Organ donor identification for each recipient was gathered through the electronic database of the Regional Transplant Coordination Centre (Regional Transplant Service, Piedmont).

### 2.1. Definitions

Enteric colonization with CP-*Kp* was defined as a patient having at least one rectal swab (RS) positive for CP-*Kp* during their hospital stay before and/or after transplantation; CP-*Kp* invasive infection was defined by clinical and laboratory evidence of any invasive infection plus at least one culture positive for CP-*Kp* from a biological sample other than RS; bloodstream infection (BSI) was defined as a patient having at least one positive blood culture (BC) for CP-*Kp.*

Appropriate empiric treatment was defined as any antimicrobial treatment initiated empirically within 24 h from the microbiological sample collection with at least 1 active agent. Appropriate targeted treatment was defined as any antimicrobial treatment containing at least one active antimicrobial agent and initiated any time after receiving the final susceptibility test, which was performed on a bacterial isolate.

### 2.2. Microbiology

The microbiological testing included the following: RS, that was performed by inoculation on MacConkey II Agar (Becton, Dickinson and Company, Sparks, MD, USA), with the automated WASP^®^ system (Copan, Brescia, Italy) followed by biochemical identification of isolated colonies and an antimicrobial sensitivity test with the MicroScan system (Beckman Coulter, Brea, CA, USA). Surveillance RSs for MDR bacteria were performed at admission and weekly on all high-risk patients (defined as patients admitted to the ICU and immunocompromised patients) according to the local Infection Control Policy (Regione Piemonte n° 30335/DB.2001). As clinically indicated, peripheral and central BCs and other samples were taken from urine, surgical wounds, drainage, tracheal bronchoaspiration (BA), alveolar bronchoalveolar lavage (BAL), bile, stools, and organ preservation solution. Bacterial growth in blood was detected by the BacT/ALERT system (bioMérieux) after 37 °C incubation with CO_2_ colorimetric detection, followed by Gram staining and bacterioscopy. Antimicrobial susceptibility was tested with the MicroScan system and defined according to the EUCAST breakpoint definitions [9]. The Hodge test and boronic acid tests on carbapenem-susceptible E. coli strains (ATCC 25922) were performed to confirm the phenotype of reduced carbapenem-susceptibility *Kp* isolates. Furthermore, detection of carbapenem resistance genes was carried out using an Xpert Carba-R assay (Cepheid, Sunnyvale, CA, USA)

### 2.3. Statistical Analysis

The primary outcomes included a description of the association of enteric colonization and invasive infections by CP-Kp with one-year mortality. Secondary outcomes were the study of risk factors for colonization and invasive infections by CP-*Kp*.

The continuous variables’ values are described as either mean and standard deviation (SD) or with a median and inter-quartile range (IQR). The categorical variables’ values are expressed as frequency measure and percentage. To compare the continuous variables, we used the t-test for normal distribution and the Wilcoxon–Mann–Whitney test for non-normal distribution. To compare discrete variables, we used the chi-square test or Fisher’s test accordingly. Considering the outcome of CP-*Kp* BSI and one-year mortality, significant values in the univariate analysis were evaluated with a multivariate model: a logistic regression model for CP-*Kp* BSI and a proportional risk model (Cox’s model) for one-year mortality. A *p*-value of <0.05 was considered significant. The statistical analysis was conducted using SAS system 9.3 (SAS Institute Inc., Cary, NC, USA).

Ethical committee approval was waived due to the retrospective nature of the study, which was approved by the medical director of the hospital (protocol number 0116265).

## 3. Results

During the study period, 920 SOTs were performed. In total, 92 were excluded from the analysis, with 16 emergency re-transplants during the same hospitalization considered a single episode, 20 pediatric recipients transferred to a different hospital for follow-up after the intervention, and 56 episodes excluded because the clinical documentation was not fully available for review. Combined transplants were considered a single episode for each organ. The final analysis included 828 episodes. The demographic features of the study population, comorbidities, hospitalization details, and ICU-associated procedures are summarized in Table 1.

Enteric colonization with CP-Kp was found in 45 patients (5.4%), with most cases occurring (88.9%) after SOT (median 12 days, IQR 4;26). There were 35 invasive infections (4.2%) with a mortality of 51.4%. These include18 (51.4%) were associated with a BSI and mortality of 66.7%. Table 2 shows the main characteristics of patients who were colonized or infected.

Among the risk factors for the development of a CP-*Kp* BSI there were CP-*Kp* enteric colonization and multi-site CP-*Kp* colonization. The duration of hospital and ICU stays were also associated with the development of a CP-*Kp* BSI as well as previous hospital admission and corticosteroid use in the six months preceding the SOT. A lung or heart transplant procedure was significantly associated with the development of a CP-*Kp* BSI (*p* < 0.0001 and *p* = 0.0057, respectively). Among the comorbidities, COPD was correlated with the development of a CP-*Kp* BSI.

Among patients with a CP-*Kp* BSI, 29,4% received appropriate empiric treatment within 24 h from the BC collection, and 88.2% received appropriate targeted treatment. While two patients with a CP-*Kp* BSI (11.1%) died of septic shock within 24 h of diagnosis, two other patients developed a CP-*Kp* BSI and were successfully treated with an appropriate empiric gentamycin monotherapy within 24 h. In addition, 13 patients (86.7%) received a combination therapy: 2 with 2 active agents, and the remaining 11 received 3 active agents. The CP-*Kp* strains that were isolated were colistin-resistant in 27% of the cases, resistant or intermediate to tigecycline in 11% and 33%, respectively, and intermediate to gentamycin in 22%, without any isolate resistance to the latter.

### Univariate and Multivariate Analyses for One-Year Mortality

The results of the univariate and multivariate analyses for one-year mortality are given in Table 3.

In the univariate analysis, colonization, invasive infections as well as sepsis, severe sepsis, and septic shock were significantly associated with one-year mortality. In the multivariate analysis, only invasive infections and the combination of sepsis, severe sepsis, and septic shock were significantly associated with one-year mortality, whereas gastrointestinal colonization was significantly associated with survival.

The Kaplan–Meier curve showed that a one-year survival rate for SOT recipients with a CP-*Kp* BSI was significantly reduced compared to those without a CP-*Kp* BSI (Figure 1).

## 4. Discussion

Over the last decade, CP-Kp has become a major challenge, particularly in Southern Europe [10]. The European burden of disease associated with infections owing to antibiotic-resistant bacteria was very high as far as Italy was concerned, with 10,762 deaths in 2015 and CP-Kp mortality ranging between 30% and 70% [11]. The European Centre for Disease Control (ECDC) reported that, in Europe, 7.5% of Klebsiella pneumoniae isolated from blood cultures and cerebral spinal fluid were resistant to carbapenems, while in Italy, it was 26.8% [12].

The aim of this study was to assess the impact of CP-*Kp* enteric colonization and invasive infection on one-year mortality after SOT in a single center with a low rate of gastrointestinal colonization by CP-*Kp,* which was analyzed with dynamic of acquisition after SOT. The inherent susceptibility to CP-*Kp* colonization and infection of an SOT recipient in the first month after transplantation is in agreement with previous reports and can be related to prolonged hospital stays, especially in a tertiary care setting where the risk of CP *Enterobacteriaceae* (CPE) circulation remains high according to the local epidemiology and other factors [13].

In a recent retrospective study, the stool sample positivity for vancomycin-resistant Enterococci and/or *Clostridioides difficile* during the 30 days prior to an SOT and an intensive care unit stay >2 weeks were associated with the acquisition of CPE, suggesting a central role played by disrupted gastrointestinal flora in the susceptibility to CPE [14]. In our study, 60% of non-CP-*Kp* BSIs were caused by pathogens that were potentially of enteric origin, highlighting the importance of the gut as a bacterial reservoir in these patients [15]. Interestingly, Giacobbe and colleagues reported that previous BSIs due to other pathogens were associated with an increased risk of a CP-Kp BSI that was independent of other factors in colonized patients with prolonged hospital exposure [16].

In our study, there was a lower colonization rate compared to a previously reported multicentre Italian study where the enteric colonization rate was 18.4% [17]. The overall CP-*Kp* invasive infection (4.2%) and CP-*Kp* BSI mortality rates (66.7%) were similar to previously published studies.

Notably, 16.8% of the studied patients did not undergo an RS during their stay, despite the regional Infection Control Policy protocol, which had already been operative in the hospital during the study period. Consequently, in 5.7% of the CP-*Kp* invasive infections and in 11.1% of the CP-*Kp* BSI cases in our cohort, the enteric colonization with MDR pathogen status was unknown, resulting in delayed administration of the appropriate antibiotic therapy and an increased risk of nosocomial spread of CP-*Kp*.

The results may appear somehow conflicting, although they should be analyzed in the context of the local epidemiology of drug resistance, immunosuppression, and the SOT population. In our study, CP-*Kp* invasive infection during the year after LT was the strongest predictor of one-year mortality; patients with a CP-*Kp* BSI had lower survival compared to those without a BSI, whereas a BSI that was caused by bacteria other than CP-*Kp* had no impact on one-year mortality. The high rate of septic shock among the deceased patients points to the difficulty of controlling these infections, which is partially due to the limited therapeutic tools available against this pathogen [18].

Even though enteric colonization was a significant risk factor for the development of CP-*Kp* invasive infection in the multivariate analysis, it was not associated with one-year mortality. There may be a multitude of explanations for this, including possible early appropriate empiric antibiotic treatment or that, in our patient population, the main drivers of invasive infections were different and possibly multiple in determining the rate of invasive infections (e.g., effective source control, the amount of bacteria in colonized patients, the strains and any effect of previous or concomitant antibiotic treatment as well as type, duration, and complications of surgeries).

The issue of mortality in patients with infections by CP-*Kp* was also studied in a large Italian multicentre study, which was not limited to SOT recipients, where Tumbarello et al. highlighted how 67% of all documented CP-*Kp* infections were associated with a BSI and had a 14-day mortality of 34% [19]. In a smaller case series of cardiac ICU patients by Sbrana et al., a mortality rate of 83% in patients with CP-*Kp* was found. All the patients had enteric colonization and received appropriate empiric antimicrobial treatment within 24 h of their clinical diagnosis [20].

Several studies have described the impact of these bacteria in different cohorts of SOT patients, highlighting an increased susceptibility to CP-*Kp* infection. A Greek study from Mouloudi et al. reported significantly higher mortality in liver transplant recipients in the ICU with a CP-*Kp* BSI compared to a control group without a CP-*Kp* BSI (82% and 32%, respectively) [21]. Similarly, Lübbert et al. reported higher infection rates and excess mortality among patients admitted to the liver transplant unit during a CP-*Kp* outbreak in a German hospital [22]. In a large Brazilian cohort of over a thousand kidney transplant recipients, 2.1% of the patients had CP-*Kp* infections, which were associated with a BSI in 38.7% of the cases, including a significantly reduced survival rate and an association with a higher SOFA score at the time of diagnosis [23].

According to a Brazilian retrospective study, kidney transplant recipients with CP-*Kp* urinary tract infections had a lower 2-year survival rate compared with those without CP-*Kp* infections (76% vs. 90%). Additionally, CP-*Kp* bacteriuria after kidney transplant was associated with mortality and antimicrobial failure after treatment [24].

In accordance with updated Italian recommendations, CPE colonization should not be viewed as an absolute contraindication for organ donation unless the colonization is in the organ to be transplanted, given the low risk of CPE transmission from donor to recipient [25].

By contrast, the role of immunosuppressive therapy is unclear in this context. Our results do not allow any deduction about the role of immunosuppressive treatment in terms of the risk of MDR pathogen infections, which has not been confirmed by previous studies [26,27].

Similarly, we cannot confirm any protective role of these drugs due to their impact on the systemic inflammatory response, as previously proposed [16,17].

Recently, novel beta-lactams/beta-lactams inhibitors combinations such as ceftazidime/avibactam (C/A), meropenem/vaborbactam (M/V) and imipenem/relebactam (I/R) significantly reduced mortality patients with carbapenem-resistant Enterobacteriaceae infections [27,28,29]. Notably, C/A is an intravenous combination of a third-generation cephalosporin with the non-β-lactam/β-lactamase inhibitor avibactam, with activity against ESBL-producing bacteria, P. aeruginosa, and KPC or OXA-48 carbapenemase-producing bacteria [28,29]. Sun et al. reported that C/A significantly reduced short-term mortality among SOT recipients with CPE infections compared with salvage regimens [30]. Data in SOT per M/V and I/R, for now, are few, and we are waiting for further studies [27].

This study has several limitations, including its retrospective nature. To ensure a better understanding of the true impact of CP-*Kp* infections on the SOT recipient subgroup, a comparison with the epidemiology of this pathogen in the entire hospital is needed. Such measurement is currently unavailable due to the lack of surveillance data about CP-Kp prevalence in the hospital wards that are not defined as high risk for MDR pathogens.

Furthermore, the data presented pertain only to the hospital admission that is related to the transplantation for each patient. A longer clinical and microbiological follow-up would be helpful for understanding the long-term impact of CP-*Kp*. Finally, the limited number of CP-Kp BSI cases did not allow any statistically significant conclusion on the use of antimicrobial agents in our cohort of patients.

## 5. Conclusions

In summary, post-transplant BSIs occurred at a high rate during the year after SOT, particularly within the first four weeks following surgery. Patients with a BSI had a significantly lower survival rate than those without a BSI. New antibiotics have recently been approved for CRE infections.

Our results suggest that identification of enteric colonization is crucial to preventing the nosocomial spread of these pathogens and protecting more vulnerable patients by safeguarding the outcome of complex and costly life-saving treatments.

## Figures and Tables

**Figure 1 microorganisms-09-02272-f001:**
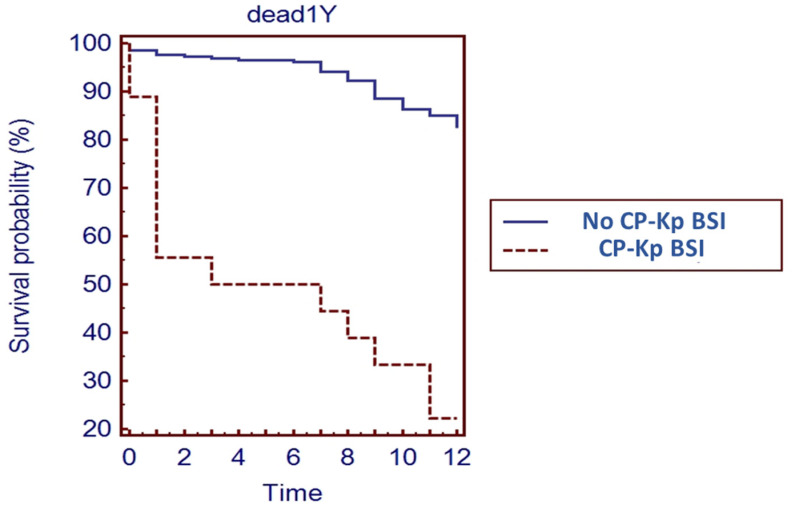
Kaplan–Meier survival curve for SOT recipients with CP-*Kp* BSI.

**Table 1 microorganisms-09-02272-t001:** Demographics, pre-transplant co-morbidities, transplant distribution, and post-transplant admission data.

Characteristics of Population	n. (%)
AGE (mean (SD))	53.8 (11.6)
MALE SEX (n. (%))	574 (69.3)
COMORBIDITIES (n. (%))	
Diabetes	177 (21.4)
Cardiovascular diseases	448 (54.1)
COPD	67 (8.1)
Chronic liver disease	371 (44.8)
Chronic renal disease	400 (48.3)
Previous solid malignancy	224 (27.1)
Previous hematologic malingancy	4 (0.5)
Splenectomy	7 (0.9)
ADMISSION WITHIN 6 MONTH PRE-TRANSPLANT (n. (%))	270 (32.6)
PREVIOUS SOT (n. (%))	
Same organ	57 (6.9)
Other organ	4 (0.5)
Time from previous transplant (median years (IQR))	9.9 (4.5; 14.9)
POSITIVE CMV IgG (n. (%))	710 (85.8)
CORTICOSTEROID TREATMENT PRE-TRANSPLANT (n. (%))	78 (9.42)
Other immunosuppressive treatment (n. (%))	7 (0.9)
BACTERIAL ISOLATES 3 MONTHS PRE-TRANSPLANT (n. (%))	93 (11.2)
ESBL positive	22 (2.7)
Carbapenemases positive	13 (1.6)
TRANSPLANTED ORGAN n. (%)	
Liver	340 (41.1)
Kidney	339 (41)
Lung	66 (7.9)
Heart	57 (6.9)
Kidney-pancreas	10 (1.2)
Kidney-liver	9 (1.1)
Split liver	5 (0.6)
Heart-lung	1 (0.1)
Lung-Liver	1 (0.1)
Total	828 (100)
OVERALL MORTALITY (n. (%))	72 (8.7)
At 30 days	26 (3.4)
At 1 year	53 (6.4)
TOTAL DURATION OF ADMISSION (median days (IQR))	19 (13; 30)
ICU STAY (median days (IQR))	7 (4; 12)
COMBINED TRANSPLANT (n. (%))	21 (2.5)
COMPLICATED TRANSPLANT (n. (%))	213 (25.7)
Re-intervention	107 (13)
Dialisis	100 (12.1)
ECMO	38 (4.6)
Re-transplant	17 (2.1)
Time to re.transplant (median days (IQR))	5 (4; 14)
SEPSIS (n. (%))	86 (10.4)
Severe sepsis	37 (4.5)
Septic shock	26 (3.1)
GRAFT LOSS (n. (%))	53 (6.4)
Time from transplant (median days (IQR))	47 (4; 340)
POST-TRANSPLANT CMV DNA (n. (%))	72 (8.7)
GANCICLOVIR TREATMENT (n. (%))	99 (12)
TRANSPLANT REJECTION (n. (%))	83 (10)
Bolus doses of steroids	84 (10.1)
Other anti-rejection therapy	8 (1)
Time from transplant (median days (IQR))	10 (6; 20)

Abbreviations: COPD: chronic obstructive pulmonary disease, CMV: cytomegalovirus, ESBL: extended-spectrum beta-lactamases, ECMO: extracorporeal membrane oxygenation, n.: number.

**Table 2 microorganisms-09-02272-t002:** Characteristics and outcomes of enteric colonization, organ infection, and bloodstream infection (BSI).

Variable	n. (%)
CP-*Kp* ENTERIC COLONIZATION	45 (5.4)
Pre-transplant	5 (0.6)
Post-transplant	40 (4.8)
Timing of colonization (median days (IQR))	12 (4; 26)
Unknown	139 (16.8)
COLONIZATION WITH OTHER CP BACTERIA	5 (0.6)
CP-*Kp* INVASIVE INFECTION	35 (4.2)
With known enteric colonization	24 (2.9 OP; 68.6 I)
Without enteric colonization	9 (1.1 OP; 25.7 I)
Unknown	2 (0.3 OP; 5.7 I)
CP-*Kp* INVASIVE INFECTION MORTALITY	18 (2.2 OP; 51.4 I)
CP-*Kp* BSI	18 (2.2)
Timing of onset after Tx (median days (IQR))	12 (8; 24)
With known enteric colonization	13 (1.6 OP; 72.2 CP-*Kp* BSI)
Without colonization	3 (0.4 OP; 16.7 CP-*Kp* BSI)
Unknown	2 (0.2 OP; 11.1 CP-*Kp* BSI)
CP-*Kp* BSI MORTALITY	12 (1.5 OP; 66.7 CP-*Kp* BSI)
At 30 days	6 (0.7 OP; 33.4 CP-*Kp* BSI)
At 1 year	11 (1.3 OP; 61.1 CP-*Kp* BSI)
SEPSIS	18 (100)
Severe sepsis	14 (1.7 OP; 77.8 CP-*Kp* BSI)
Septic shock	9 (1.1 OP; 50 CP-*Kp* BSI)
POSSIBLE SOURCE FOR CP-*Kp* BSI	
Abdomen	2 (11.1)
Urine	2 (11.1)
Respiratory system	8 (44.5)
SSTIs	1 (5.6)
CVC-related	1 (5.6)
Organ preservation solutions	1 (5.6)
Unknown origin	3 (16.7)

Abbreviations: CP-*Kp*: carbapenemase-producing *Klebsiella pneumoniae*, OP: overall population, I: CP-*Kp* organ infection, BSI: bloodstream infection, Tx: transplant, CVC-related: central venous catheter-related.

**Table 3 microorganisms-09-02272-t003:** Univariate and multivariate analysis for risk factors for 1-year mortality.

	Univariate Analysis			Multivariate Analysis (Cox’s Model)			
Variable (n. (%))	Dead at 1 y	Alive at 1 y	*p*	HR	CI 95%		*p*
Enteric CP-*Kp*	14 (31.1)	31 (6.1)	<0.0001	0.628	0.462	0.855	0.0031
Multi-site CP-*Kp*	7 (13.7)	5 (0.7)	<0.0001	0.606	0.196	1.877	0.3851
CP-*Kp* organ infection	16 (30.8)	19 (2.5)	<0.0001	2.533	1.053	6.094	0.0380
CP-*Kp* BSI	11 (21.6)	7 (0.9)	<0.0001	2.511	0.808	7.803	0.1116
Sepsis	30 (58.8)	58 (2)	<0.0001	2.421	1.536	3.815	0.0001
Severe sepsis	23 (45.1)	14 (1.8)					
Septic shock	20 (39.2)	6 (0.8)					

Abbreviations: n.: number.

## Data Availability

The data presented in this study are available on request from the corresponding author.

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
