# Peer review of "Carbapenemase-Producing Klebsiella pneumoniae Colonization and Infection in Solid Organ Transplant Recipients: A Single-Center, Retrospective Study"

_microorganisms, 2021, doi:10.3390/microorganisms9112272_

Round 1

Reviewer 1 Report

This a retrospective single center study of colonization and infection due to carbapenemase producing K. pneumonia isolates in solid transplant organ recipients. The study is descriptive.

Mayor points.

Result were obtained from May 2011 to August 2014. Why so delay in communicating the results?  

The microbiological methods should be further described. Authors explained how rectal swabs were seeded but not which culture media were used. Where molecular methods also used? The Hodge test is no longer recommended. Do the authors used other alternative methods to confirm the carbapenemase production?

Microorganisms should be in italics.

Specific points

Line 111. MicroScan is no longer Siemmens   

Table 2. Transplant instead tranplant

Figure 1. Why KPC? In my understanding molecular methods to characterize carabapenemase enzymes were not used. Is cero time when transplantation was performed?  

Author Response

Dear Editor,

Thanks for the opportunity to revise our manuscript. Please find the response to reviewers’ comments on our manuscript entitled “Carbapenemase-producing Klebsiella pneumoniae Colonization and Infection in Solid Organ Transplant Recipients: a Single Center, Retrospective Studyto be considered for publication in Microorganisms.

Reviewer comments

Reviewer  1#

This a retrospective single center study of colonization and infection due to carbapenemase producing K. pneumonia isolates in solid transplant organ recipients. The study is descriptive.

Mayor points.

  1. Result were obtained from May 2011 to August 2014. Why so delay in communicating the results?

Dear reviewer, thank you for this comment. We have recently taken over the database which had been set aside for other projects, subsequently, the onset of the covid pandemic further delayed the production of labour. We hope that this manuscript will be of interest to you.

  1. The microbiological methods should be further described. Authors explained how rectal swabs were seeded but not which culture media were used. Where molecular methods also used? The Hodge test is no longer recommended. Do the authors used other alternative methods to confirm the carbapenemase production?

Dear reviewer thank you for this comment. We have completed the methods section adding alternative tests used to confirm the carbapenemase production. Detection of carbapenem resistance genes was carried out using an Xpert Carba-R assay (Cepheid, Sunnyvale, CA, USA)

  1. Microorganisms should be in italics.

Dear reviewer thank you for this comment. The names of microorganisms have been revised according to your suggestions throughout the text.

Specific points

  1. Line 111. MicroScan is no longer Siemmens  

Dear reviewer, thank you for this comment. We have changed siemens into Beckman Coulter 

  1. Table 2. Transplant instead tranplant

Thank you. Table 2 has been revised accordingly to your suggestions

  1. Figure 1. Why KPC? In my understanding molecular methods to characterize carabapenemase enzymes were not used. Is cero time when transplantation was performed?  

Dear reviewer, thank you for this comment. We agree with you. KPC is an error that occurred in the figure caption. We have changed figure 1 according to your suggestions

Reviewer 2 Report

In this study, Pagani et al evaluated the one-year mortality and risk factors for colonization and infections by CR-Kp. Overall, results from univariate and multivariate analyses revealed that the 1-year mortality was associated with invasive infections, unlike gastrointestinal colonization. Although the topic is of interest, the overall novelty of results appears limited.

Major comments

- It could be worth mentioning which type of carbapenemase was produced by CR-Kp included in this study

- While discussing the overall mortality rates observed in SOT patients, please note that such rates appear to be drastically reduced following the introduction of novel BLBLICs to treat CRE infections. A comment about this point would be welcomed in the Discussion section (e.g. l.238-158) (see for example 10.1093/ofid/ofz360.2345)

Minor Comments

l.54: please consider rephrasing

l.27: the CP-

l.64-65: please provide reference, and examples of cited new antibiotics

l.109: details about used selective media are welcomed

l.110: provide more details about biochemical identification

l.120: BP version?

l.176: please specify more details about used agents

l.202 I would add a comment about the epidemiology of CPE in your country in comparison to that observed at local scale

l.204: Clostridioides

l.206: grastrointestinal microbiota

Author Response

Dear Editor,

Thanks for the opportunity to revise our manuscript. Please find the response to reviewers’ comments on our manuscript entitled “Carbapenemase-producing Klebsiella pneumoniae Colonization and Infection in Solid Organ Transplant Recipients: a Single Center, Retrospective Studyto be considered for publication in Microorganisms.

Reviewer #2:

In this study, Pagani et al evaluated the one-year mortality and risk factors for colonization and infections by CR-Kp. Overall, results from univariate and multivariate analyses revealed that the 1-year mortality was associated with invasive infections, unlike gastrointestinal colonization. Although the topic is of interest, the overall novelty of results appears limited.

Major comments

- It could be worth mentioning which type of carbapenemase was produced by CR-Kp included in this study

Dear reviewer thank you for this comment. This retrospective study included patients from 2011 to 2014 with colonization or infection due to carbapenemases producers Kp. The Hodge Test and boronic acid tests on carbapenem-susceptible E. coli strains (ATCC 25922) were performed to confirm the phenotype of reduced carbapenem-susceptibility Kp isolates. Furthermore, detection of carbapenem resistance genes was carried out using an Xpert Carba-R assay (Cepheid, Sunnyvale, CA, USA). Despite that, in the period of study, we cannot assess which type of carbapenemase was produced by CR-Kp included in this study

While discussing the overall mortality rates observed in SOT patients, please note that such rates appear to be drastically reduced following the introduction of novel BLBLICs to treat CRE infections. A comment about this point would be welcomed in the Discussion section (e.g. l.238-158) (see for example 10.1093/ofid/ofz360.2345)

Dear reviewer, thank you for this comment that improves notably our manuscript. We agree with you. We have added a paragraph regarding Novel BL/BLI combinations and data available on SOTs

Minor Comments

l.54: please consider rephrasing

Dear reviewer, thank you for this comment. We have changed the text according to your suggestions

l.27: the CP-

Dear reviewer, thank you for this comment. We have changed the text according to your suggestions

l.64-65: please provide reference, and examples of cited new antibiotics

Dear reviewer, thank you for this comment. We have revised the text and rephrased this sentence to avoid misunderstanding

l.109: details about used selective media are welcomed

Dear reviewer, thank you for this comment. We have improved the methods section according to your suggestions. You can find changes highlighted in the text.

l.110: provide more details about biochemical identification

Dear reviewer, thank you for this comment. We have improved methods and biochemical identification sections according to your suggestions. You can find changes highlighted in the text.

l.120: BP version?

Dear reviewer, thank you for this comment. We have improved the method section according to your suggestions. You can find changes highlighted in the text.

l.176: please specify more details about used agents

Dear reviewer, thank you for this comment. Data showed in the manuscript contains all data available regarding used agents. Unfortunately, we have no further data.

l.202 I would add a comment about the epidemiology of CPE in your country in comparison to that observed at local scale

Dear reviewer, thank you for this comment. We have added in the text new data on CP-Kp incidence and mortality in Europe and Italy. Comment and comparisons between our data and previous data were done throughout the text.

l.204: Clostridioides

Dear reviewer thank you for this comment. The text has been revised accordingly to your suggestion

l.206: grastrointestinal microbiota

Dear reviewer thank you for this comment. The text has been revised accordingly to your suggestion